# WHY IS THE LLM UNSURE? PROFILING THE CAUSES OF LLM UNCERTAINTY FOR ADAPTIVE MODEL AND UNCERTAINTY METRIC SELECTION

## ABSTRACT

Large Language Models (LLMs) frequently produce fluent yet factually inaccurate outputs, termed hallucinations, which compromise their reliability in real-world applications. Although uncertainty estimation offers a promising approach to detect these errors, existing metrics lack interpretability and offer limited insight into the underlying causes of uncertainty. In this work, we introduce a novel prompting-based framework for systematically analyzing the causes of uncertainty in LLM responses. We design dedicated indicators to quantify each distinct cause and profile how existing uncertainty metrics align with them. Our findings reveal systematic variations in uncertainty characteristics across metrics, tasks, and models. Leveraging these insights, we propose a task-specific metrics/models selection method guided by the alignment of uncertainty characteristics with task requirements. Experiments across multiple datasets and models demonstrate that our selection strategy consistently outperforms non-adaptive baselines, achieving 3-4% performance improvements and enabling more reliable and efficient uncertainty estimation for LLM deployment. Code is available at link.

## 1 INTRODUCTION

Large language models (LLMs) have demonstrated remarkable capabilities, often surpassing average human performance on tasks involving mathematics, reasoning, and programming. Despite these advances, LLMs frequently produce confident yet factually incorrect responses, commonly referred to as *hallucinations*.

A promising direction for mitigating hallucinations lies in *uncertainty estimation*. Prior work has proposed a variety of metrics to flag unreliable outputs, including distribution-based, verbalized confidence, and perturbation methods (Malinin & Gales, 2020; Kadavath et al., 2022; Gao et al., 2024). Despite their effectiveness at detecting uncertainty, they share a common limitation: ***they offer limited insight into the underlying causes of uncertainty***. Most metrics return a single score indicating "how uncertain" the model is, without revealing *why* the model is uncertain. This lack of interpretability makes it difficult to understand model behavior or design targeted interventions to improve response reliability. For example, is the model uncertain due to an ambiguous question, insufficient knowledge, or instability under reflection? Without such distinctions, designing effective corrective strategies is difficult.

Motivated by this gap, we address two research questions in this paper:

- Can we design interpretable indicators that quantify distinct causes of uncertainty and suggest actionable interventions?
- Given such indicators, can we use them to adaptively select suitable models or uncertainty metrics for a given task?

To tackle the first question (Section 3 & 4), we propose a prompting-based framework that links uncertainty estimation with interventions shown to improve response correctness (Zhou et al., 2024; Shinn et al., 2023). We attribute LLM uncertainty to four distinct causes: *syntax sensitivity*, *semantic ambiguity*, *indecisiveness among outputs*, *unconfidence when challenged* and design dedicated

indicators for each. These causes align naturally with known prompting strategies: paraphrasing reduces syntax sensitivity, clarification resolves semantic ambiguity, answer aggregation/ensembling mitigates indecisiveness among outputs, and self-reflection stabilizes challenged responses. By making these links explicit, our framework moves uncertainty estimation beyond hallucination detection toward interpretable diagnosis, explaining both *when* a response is unreliable and *why*.

Having established this framework, we then address the second question (Sections 5& 6). We use the cause indicators to construct structured 4-dimensional *uncertainty profiles* that capture how different tasks, models, and existing uncertainty metrics vary in their sensitivity to each cause. Our analysis reveals that uncertainty metrics differ systematically in the causes they capture, tasks exhibit distinct uncertainty patterns, and models display strengths and weaknesses along these axes.

Building on these insights, we propose a simple adaptive selection method that uses *uncertainty profiles* to guide the choice of uncertainty metric or model for a given task (Section 6). Specifically, we represent datasets, models, and metrics as profile vectors and select those whose uncertainty behavior best complements the task. This approach enables *task-aware* and *cause-aware* decision-making, avoiding one-size-fits-all heuristics.

Experiments across diverse models, datasets, and uncertainty metrics show that our adaptive strategy consistently outperforms non-adaptive baselines in three practical scenarios: metric selection, model selection, and joint selection. These results highlight the value of understanding the structure of LLM uncertainty for more reliable and efficient deployment.

The major contributions of this paper are as follows:

- We propose a novel framework that decomposes LLM uncertainty into four interpretable causes, represented as an *uncertainty profile* capturing the fine-grained structure of model uncertainty at the response level.

- We conduct a comprehensive analysis of *uncertainty profiles* across a range of metrics, tasks, and model families, revealing consistent and interpretable patterns in how they exhibit and respond to different causes of uncertainty.

- We introduce an adaptive model/metric selection method guided by *uncertainty profiles*, achieving consistent 3–4% NDCG improvements over non-adaptive baselines across multiple real-world scenarios.

## 2 RELATED WORK

### 2.1 UNCERTAINTY ESTIMATION FOR LLMS

Uncertainty estimation for large language models (LLMs) has been studied through a range of complementary approaches. *Generation likelihood*-based metrics quantify predictive uncertainty from the entropy of the output distribution. Examples include Normalized Predictive Entropy (NPE) and Length-Normalized Predictive Entropy (LNPE), which normalize entropy either across tokens or sequence length to mitigate length bias (Malinin & Gales, 2020), and Semantic Entropy (SE), which clusters semantically similar responses and computes entropy over these clusters (Kuhn et al., 2023). *Verbalized*-based measures prompt the model to state its own confidence directly; for instance, Verbalized Confidence (VC) and P(True) query the model for confidence statements or its probability of generating the token "True" (Kadavath et al., 2022). Beyond distributional likelihood and self-reflection, *lexical consistency-based* metrics measure stability across multiple generations, such as by computing lexical overlap scores (e.g., ROUGE) among sampled outputs (Lin et al., 2022). Finally, some methods explicitly target the decomposition of epistemic versus aleatoric uncertainty through input perturbations or conditional prompting, as in SPUQ (Gao et al., 2024) and IPT-EU (Yadkori et al., 2024).

### 2.2 PROMPTING TECHNIQUES FOR RELIABILITY

Prompting techniques have been widely explored as lightweight interventions to improve the accuracy and reliability of large language models. Paraphrasing has been shown to mitigate sensitivity to superficial wording changes and improve robustness across tasks (Zhou et al., 2024; Fu et al.,

2024; Liu et al., 2024). Clarification prompting helps resolve semantic ambiguity by encouraging the model to interpret underspecified inputs more precisely (Hou et al., 2023; Gao et al., 2024). Self-reflection and iterative refinement methods (Shinn et al., 2023; Renze & Guven, 2024) prompt models to critique and revise their outputs, reducing errors in reasoning-intensive tasks. While these prompting-based methods were originally proposed as heuristics for boosting accuracy, they can also be seen as interventions that align with specific causes of LLM uncertainty. Our work formalizes this connection by decomposing uncertainty into interpretable causes and showing how prompting strategies interact with them.

## 3 PROFILING CAUSES OF LLM UNCERTAINTY: A LITERATURE-GROUNDED TAXONOMY

LLM uncertainty stems from a variety of underlying causes. In this section, we propose a taxonomy of four primary *causes* of uncertainty in LLM responses. Grounded in prior research on calibration, uncertainty quantification, and model failure modes, this taxonomy highlights distinct, interpretable patterns of uncertainty that frequently arise in practice. We define the four possible causes of LLM uncertainty as follows:

**Syntax Sensitivity (SS)** occurs when the model struggles to interpret the linguistic surface form of a prompt—due to rare vocabulary, complex grammar, or unfamiliar phrasing—leading to shallow comprehension or misinterpretation. Prior work shows that even minor changes in syntax can significantly shift model behavior. For example, (Zhou et al., 2024) demonstrate that slight rewordings in math problems can lead to large variations in answer accuracy, highlighting the sensitivity of LLMs to input wording.

**Semantic Ambiguity (SA)** arises from inherent vagueness or under-specification in the input. Even with clear syntax, prompts may admit multiple plausible interpretations, often leading to diverse or inconsistent responses. This is especially common in open-ended or instruction-following tasks. Prior work (Hou et al., 2023; Gao et al., 2024) explores methods to identify and mitigate such ambiguity using prompt clarification and perturbation strategies.

**Indecisiveness among Outputs (IO)** arises when LLM struggles to commit to a single answer, producing inconsistent or contradictory outputs. This indecisiveness often reflects underlying gaps in training data, limited world knowledge, or difficulties with multi-step reasoning. Prior studies such as (Yadkori et al., 2024), (Ling et al., 2024), and (Ahdritz et al., 2024) explore methods for detecting and isolating such uncertainty, particularly focusing on the boundary between what models confidently know and what they cannot reliably infer.

**Unconfidence while being Challenged (UC)** arises when a model is prompted to reflect on or revise its previous answer. The model may either correct earlier mistakes or introduce new ones, revealing instability in its reasoning process. This reflects a fluctuation in confidence when challenged to justify or reconsider its output. Recent work by (Shinn et al., 2023) and (Renze & Guven, 2024) demonstrates how reflective prompting can expose such inconsistencies and impact the reliability of model responses.

Overall, the taxonomy outlines a set of interpretable and recurring causes of uncertainty in LLM behavior. While not exhaustive, it offers a practical lens for examining how and why LLMs produce hallucination responses. By quantifying these causes, we hope to support more targeted analyses of uncertainty and inform the development of future metrics.

## 4 ESTIMATING UNCERTAINTY CAUSES

In this section, we present a novel framework for quantifying the four primary causes of LLM uncertainty mentioned in Section 3. We also conduct empirical validation to demonstrate the effectiveness of our indicators in attributing uncertainty to specific underlying causes.

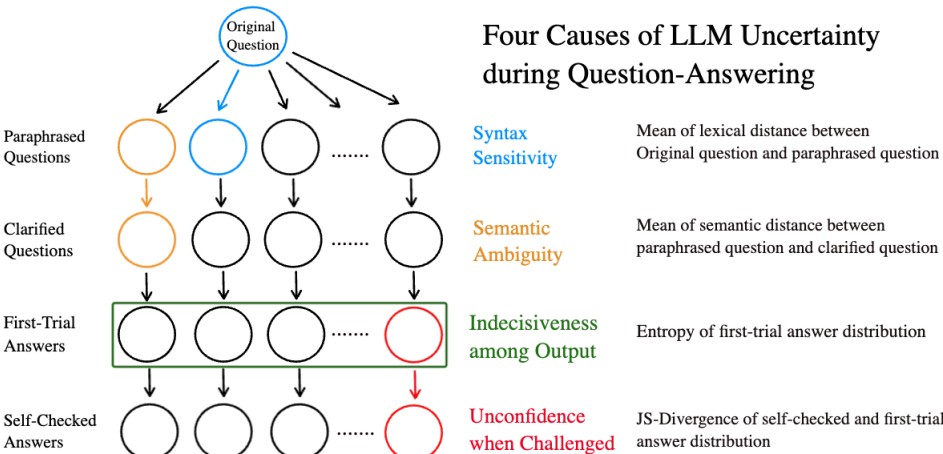

Figure 1: **Estimating Causes of LLM Uncertainty.** Our pipeline decomposes response uncertainty into four interpretable causes using a multi-stage prompting framework. Beginning with a single question, the model generates multiple response chains that proceed through paraphrasing, clarification, answering, and self-checking. Divergences introduced at each stage are used to estimate specific causes of uncertainty.

## 4.1 ESTIMATION PIPELINE

To put the uncertainty cause taxonomy into practice, we design a multi-stage prompting pipeline that reveals different manifestations of uncertainty during the question-answering process. Each stage of the pipeline is aligned with one of the four underlying *causes* of LLM uncertainty, allowing us to disentangle their distinct behavioral effects. As illustrated in Figure 1, the pipeline begins with a single input query and expands into $N$ independent response chains. Each chain progresses through four sequential stages—paraphrasing, clarification, answering, and self-checking. To quantify the contribution of each cause, we construct dedicated indicators grounded in their conceptual definitions. These indicators analyze model behavior across the response chains and capture uncertainty patterns specific to each stage of the pipeline.

**Estimating SS:** From the original question, the model generates multiple paraphrased variants that differ in lexical surface form but preserve meaning. We use these variants to estimate *Syntax Sensitivity*—the extent to which superficial changes in wording influence model behavior. Indicator $\hat{SS}$ computes the average lexical distance between the original question $q_0$ and its paraphrases $q_p^{(i)}{}_{i=1}^N$:
$\hat{SS}(q_0, \{q_p^{(i)}\}) = \frac{1}{N} \sum_{i=1}^N \text{LexDist}(q_0, q_p^{(i)})$, where $\text{LexDist}(\cdot, \cdot)$ is a lexical distance measure that is sensitive to wording differences but does not account for semantic equivalence (e.g., ROUGE). Higher values suggest that the model is more sensitive to minor lexical variations in input phrasing.

**Estimating SA:** For each paraphrased question, the model is prompted to clarify the query. The resulting semantic shift between the paraphrased and clarified versions reflects *Semantic Ambiguity*—capturing how much the model interprets the input as unclear. Indicator $\hat{SA}$ quantifies the degree of input ambiguity by computing the average embedding distance between each paraphrase $q_p^{(i)}$ and its clarification $q_c^{(i)}$: $\hat{SA}(\{q_p^{(i)}, q_c^{(i)}\}) = \frac{1}{N} \sum_{i=1}^N \text{Dist}\big(\text{vec}(q_p^{(i)}), \text{vec}(q_c^{(i)})\big)$, where Dist is a distance metric (e.g., cosine distance, L2 distance) and $\text{vec}(\cdot)$ denotes embedding extracted from the model's hidden representations. Larger values indicate greater ambiguity in the model's interpretation of the input. More details in Appendix 8.2.

**Estimating IO:** For each clarified question, the model generates an initial response. This stage captures the model's attempt to commit to a single answer based on its existing knowledge and reasoning capabilities. When the model produces inconsistent or contradictory answers across response chains, it reveals *Indecisiveness among Outputs*—a sign of underlying knowledge gaps, limited training coverage, or uncertainty in multi-step reasoning. Indicator $\hat{IO}$ quantifies this by computing the entropy of the first-trial answer distribution: $\hat{IO} = \mathcal{H}(P_{\text{FT}}) = -\sum_{a \in \mathcal{A}} P_{\text{FT}}(a) \log P_{\text{FT}}(a)$,

where $P_{\text{FT}}$ is the empirical distribution over first-trial answers. Higher entropy indicates greater uncertainty due to knowledge limitations or inconsistent reasoning.

**Estimating UC:** In the final stage, the model is asked to reflect on and potentially revise its initial response. Changes between first-trial and self-checked answers reflect *Unconfidence while being Challenged*—the model's tendency to waver or revise when prompted to reconsider its reasoning. This instability can reveal a lack of internal consistency or fragile confidence in its original output. Indicator $\hat{UC}$ quantifies this by computing the Jensen-Shannon divergence between the first-trial(FT) and self-checked(SC) answer distributions: $\hat{UC} = D_{\text{JS}}(P_{\text{SC}} \| P_{\text{FT}}) = \frac{1}{2}D_{\text{KL}}(P_{\text{SC}} \| M) + \frac{1}{2}D_{\text{KL}}(P_{\text{FT}} \| M)$. Higher values indicate lower confidence in model's reasoning when prompted to verify its response.

## 4.2 INTERPRETIVE SCOPE

***While we use the term estimate for ease of communication, our approach is better seen as generating interpretable indicators—or features—that reflect distinct behavioral tendencies linked to response uncertainty.*** Unlike black-box metrics, these features offer insight into why a model may be uncertain or incorrect, improving interpretability for downstream use. Although we strive for independence among the four features, perfect disentanglement is not possible due to interactions between uncertainty types and stage-wise propagation. Still, each feature is purposefully aligned with a distinct behavioral signature observed during prompting. In the next subsection, we show that each feature has independent predictability, reinforcing both their interpretability and practical utility for uncertainty diagnosis. Detailed prompt templates of each stage can be found in Appendix 8.8.

## 4.3 INDICATOR PREDICTABILITY VALIDATION

To validate our four indicators, we assess their ability to predict hallucinations in model responses. We generate 32 responses per question, labeling those that majority answer is wrong as *uncertain* (positive class). This allows us to evaluate how well each indicator identifies low-confidence or error-prone questions driven by specific uncertainty factors. Following (Kuhn et al., 2023; Jiang et al., 2024), we use AUROC (Area Under Receiver Operating Characteristic Curve) and AUPRC (Area Under Precision-Recall Curve) (Qi et al., 2021) as our evaluation metrics.

Evaluations are conducted across three benchmark datasets: COMMONSENSEQA (Talmor et al., 2018), GSM8K (Cobbe et al., 2021) and TRUTHFULQA (Lin et al., 2021)—using four language models from two families (Grattafiori et al., 2024; Team et al., 2024): LLAMA-3.2-3B, LLAMA-3-8B, GEMMA-2-2B, and GEMMA-2-9B. Strong performance in AUROC and AUPRC indicates that an indicator effectively captures uncertainty signals related to hallucination, supporting its use in risk assessment.

As shown in Table 1, $\hat{IO}$ consistently achieves strong AUROC scores, demonstrating its effectiveness in capturing knowledge-related uncertainty that correlates with correctness. $\hat{UC}$ also performs well, particularly on MATH, where multi-step reasoning increases the likelihood of execution errors, causing greater fluctuation in the model's answers during reflection. In contrast, $\hat{SS}$ and $\hat{SA}$ show lower performance on COMMONSENSEQA and TRUTHFULQA, likely due to the short and unambiguous nature of the questions, which limits both surface-level variation and ambiguity. Nevertheless, these indicators remain valuable for real-world scenarios involving short or ambiguous inputs. Overall, each indicator provides complementary information about different causes of uncertainty and failure modes of LLM.

## 5 PROFILING UNCERTAINTY ACROSS TASKS, MODELS, AND METRICS

Having validated our four indicators, we next analyze how the causes of uncertainty manifest across tasks, models, and standard LLM uncertainty metrics. Using the indicator pipeline introduced in the previous section, we apply our framework to all dataset–model pairs, thereby generating comprehensive uncertainty profiles. Our goal is to answer three key questions: (1) *Which types of uncertainty do existing metrics primarily capture?* (2) *Do certain tasks tend to trigger specific causes of uncertainty?* (3) *Are different language models more prone to particular types of uncertainty?*

Table 1: **Predictive Performance of Uncertainty Cause Indicators.** AUROC and AUPRC scores across three benchmarks. $\hat{IO}$ and $\hat{UC}$ consistently perform well, while $\hat{SS}$ and $\hat{SA}$ show weaker signals likely due to dataset clarity and brevity.

| Dataset | $\hat{SS}$ | $\hat{SA}$ | $\hat{IO}$ | $\hat{UC}$ |
|---|---|---|---|---|
| | AUROC / AUPRC | AUROC / AUPRC | AUROC / AUPRC | AUROC / AUPRC |
| CommonsenseQA | 0.528 / 0.346 | 0.516 / 0.322 | 0.649 / 0.444 | 0.502 / 0.335 |
| MATH | 0.544 / 0.630 | 0.437 / 0.533 | 0.763 / 0.770 | 0.777 / 0.792 |
| TruthfulQA | 0.528 / 0.358 | 0.419 / 0.290 | 0.707 / 0.516 | 0.638 / 0.471 |

## 5.1 ANALYTICAL APPROACH

To analyze how the four primary causes of uncertainty manifest across different metrics, tasks, and models, we apply our four indicators and eight benchmark uncertainty metrics to every dataset–model combination. Specifically, we experiment on four datasets (MATH, CommonsenseQA, TriviaQA, TruthfulQA) and five open-source models (LLaMA-3.2-3B, LLaMA-3-8B, Gemma-2-2B, Gemma-2-9B, Mistral-7B-v0.3), evaluating eight *task-agnostic uncertainty metrics* spanning token-likelihood, verbalized confidence, lexical consistency, and epistemic/aleatoric decomposition (see Section 2). Details of the experimental setup are provided in Appendix 8.5 and 8.7.

For each question, we compute both the cause indicators and uncertainty metrics, then average them to obtain a single score per dataset–model pair. This yields a table where each row contains four cause-indicator values and eight metric scores. By aggregating along one axis (e.g., models, tasks, or metrics), we can examine how uncertainty patterns vary across the others. The analysis proceeds in three complementary steps:

**Metric-Level Attribution** To assess which cause of uncertainty each metric primarily reflects, we average the mutual information (MI) between each benchmark metric and each cause indicator across all dataset-model pairs. MI is a general-purpose statistical measure that quantifies the dependency between two random variables, without assuming a linear relationship or specific distributional form. This makes it particularly well-suited for analyzing non-monotonic or complex associations between cause indicator values and uncertainty metric scores. Formally, MI between random variables $U$ (indicator value) and $M$ (metric score) is defined as: $I(U; M) = \sum_{u \in \mathcal{U}} \sum_{m \in \mathcal{M}} P(u, m) \log\left(\frac{P(u,m)}{P(u) P(m)}\right)$. where $P(u, m)$ is the joint probability distribution, and $P(u)$ and $P(m)$ are the marginal distributions. Higher MI values indicate stronger statistical dependence, suggesting that the metric is more sensitive to variation in that specific cause of uncertainty.

**Task-Level Attribution** To examine how uncertainty varies by task, we average each cause indicator values across all models for each dataset, yielding a task-level profile of uncertainty. This highlights which types of uncertainty are most prominent in different task settings (eg. arithmetic vs. commonsense vs. fact-checking).

**Model-Level Attribution** Third, to assess model-level tendencies, we average indicator values across all datasets for each model. Higher values indicate greater susceptibility to specific cause of uncertainty, potentially reflecting architectural biases or capacity limitations.

Together, these analyses reveal how different causes of uncertainty are distributed across tasks, models, and evaluation metrics.

## 5.2 DISCUSSION AND INSIGHTS FROM UNCERTAINTY PROFILING

**What type of uncertainty does each metric generally capture?** Figure 2 presents the mutual information between each benchmarked metric and the four uncertainty cause indicators across datasets. A dataset-wise breakdown reveals that uncertainty metrics behave differently depending on the task. For example, EU-specific metrics such as IPT-EU show significantly higher mutual information with $\hat{IO}$ on datasets like MATH and TRIVIAQA, where reasoning and factual knowledge play a central role. In contrast, metrics like VC-NEG consistently exhibit low mutual information

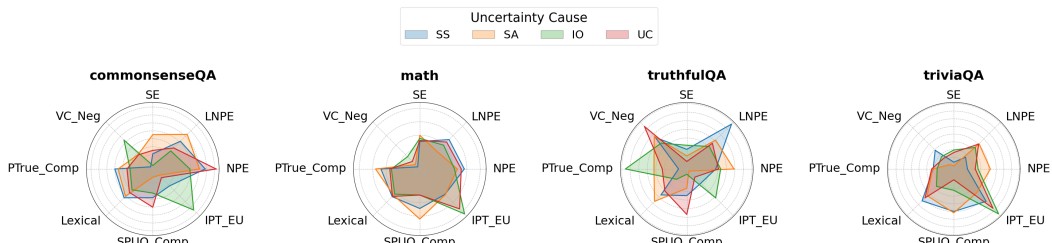

Figure 2: **Uncertainty Profiles of Existing Metrics.** The figure shows the relative magnitude of mutual information between the 4 uncertainty cause indicators and 8 benchmarked uncertainty metrics, normalized within each cause. A higher peak of MI indicates that the corresponding metric is more strongly influenced by that cause compared to other metrics.

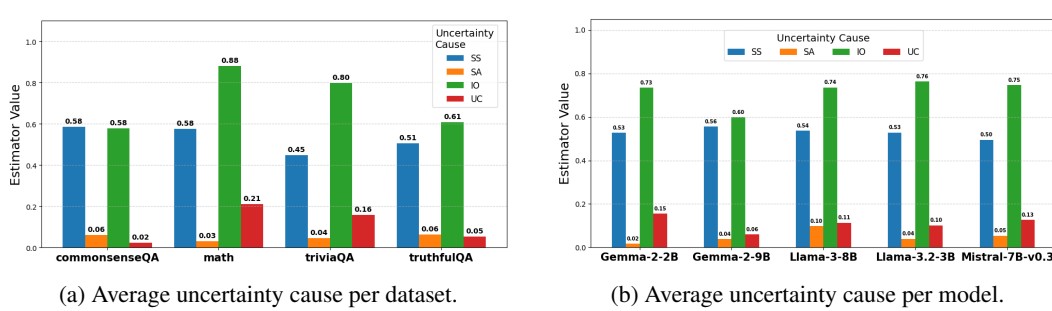

(a) Average uncertainty cause per dataset.          (b) Average uncertainty cause per model.

Figure 3: **Uncertainty Profiles of Datasets and Models.** (Left) $\hat{SS}$ and $\hat{SA}$ are relatively consistent across datasets, while $\hat{IO}$ and $\hat{UC}$ vary more significantly, particularly elevated in MATH and TRIVIAQA. (Right) Uncertainty profiles remain fairly stable across model sizes, with $\hat{SS}$ and $\hat{IO}$ being the dominant contributors, while $\hat{SA}$ and $\hat{UC}$ are generally less pronounced.

across most uncertainty causes on MATH, indicating limited capacity to capture complex reasoning-related uncertainties.

**Do certain tasks consistently exhibit stronger tendencies toward specific uncertainty cause?** Figure 3a shows the average values of each uncertainty cause indicator across datasets. Overall, $\hat{SS}$ and $\hat{IO}$ emerge as the most prominent uncertainty cause across tasks. In contrast, $\hat{UC}$ is particularly pronounced in MATH and TRIVIAQA, while $\hat{SA}$ remains consistently low across all datasets. These patterns are consistent with prior research, which has identified similar challenges related to uncertainty in language model performance. High $\hat{SS}$ scores suggest that models are sensitive to input phrasing, implying that techniques like prompt optimization (Khattab et al., 2024; Yang et al., 2023) or paraphrasing (Zhou et al., 2024) could significantly impact performance. $\hat{IO}$ tends to be higher in tasks requiring complex reasoning(MATH) or reading comprehension (TRIVIAQA). These tasks often involve open-ended or essay-style questions rather than multiple-choice formats (COMMONSENSEQA, TRUTHFULQA), which places greater demands on the model to generate accurate and complete responses from scratch (Yadkori et al., 2024). $\hat{UC}$ is also elevated in these datasets, likely because longer, multi-step generative answers are more prone to execution or formatting errors compared to shorter, constrained outputs (Xu et al., 2025; Zeng et al., 2025). In contrast, $\hat{SA}$ is relatively low across all datasets, likely because the benchmark questions are carefully curated to reduce ambiguity.

**Are different language models more susceptible to specific uncertainty cause?** Figure 3b presents the average values of each uncertainty cause indicators across models. Overall, the distribution of uncertainty sources is relatively consistent across models, with $\hat{SS}$ and $\hat{IO}$ being the dominant contributors, while $\hat{SA}$ and $\hat{UC}$ remain lower. This suggests that, regardless of model size, surface-level sensitivity and knowledge-related uncertainty are the most prominent challenges. Interestingly, there is no clear distinction between smaller and larger models in terms of their aver-

age uncertainty cause values. This contrasts with the intuitive expectation that smaller models would exhibit higher uncertainty, leading to lower accuracy. One explanation is that while uncertainty levels may be similar, smaller models are more sensitive to changes, whereas larger models tolerate them better due to stronger capabilities.

# 6 UNCERTAINTY PROFILE-GUIDED METRIC/MODEL SELECTION

In real-world applications, the performance of uncertainty metrics often exhibits significant variation between different tasks and models, presenting challenges to select the most appropriate metric or model. In this section, we propose an **adaptive selection method** that leverages uncertainty profiles to guide informed decision-making. We demonstrate its applicability in three scenarios: (1) *selecting the most suitable uncertainty metric for a given task*, (2) *selecting the most appropriate model for a given task*, and (3) *selecting the optimal uncertainty metric for a given task–model combination*.

## 6.1 ADAPTIVE SELECTION PROCESS

We represent each dataset, model, and metric as a 4-dimensional vector—*Dataset-Vec*, *Model-Vec*, and *Metric-Vec*—where each dimension corresponds to one of the four causes of uncertainty. The adaptive selection process proceeds in four steps:

1. Compute the uncertainty profiles of the target dataset, candidate models and metrics.

2. Convert each profile into a normalized 4-dimensional vector (*Dataset-Vec*, *Model-Vec*, *Metric-Vec*). Further details on vector construction and normalization are provided in Appendix 8.3 and 8.4.

3. Measure alignment between target and candidate vectors using cosine similarity.

4. Select the model or metric according to the alignment strategies (next subsection).

## 6.2 ALIGNMENT STRATEGIES

**Scenario 1** In this scenario, we aim to select the most appropriate uncertainty metric for a given dataset. We compare the dataset's *Dataset-Vec* with each metric's *Metric-Vec* and select the metric with the highest cosine similarity, as it best aligns with the task's uncertainty profile and is more likely to provide meaningful evaluation signals.

**Scenario 2** In this scenario, we select the model that is likely to perform best on the given task. Our hypothesis is that a model whose uncertainty profile is less similar to the task's (i.e., lower cosine similarity to the *Dataset-Vec*) should be a better choice, as it suggests the model is less prone to the task's dominant sources of uncertainty, leading to better performance.

**Scenario 3** In this scenario, we jointly consider the uncertainty profiles of both the task and model. For each metric, we compute its cosine similarity to both the *Dataset-Vec* and the *Model-Vec*. We then take the geometric mean of these two similarities and select the metric with the highest value. This approach favors metrics that align well with both task and model, penalizing one-sided fits.

**Evaluation** We conduct experiments across four datasets, five models, and eight uncertainty metrics, following the same configuration for metric calculation as outlined in Appendix 8.6. Each dataset is randomly split into 30% train and 60% test sets. The adaptive process is then applied to the train set to select candidate models or metrics, which are subsequently evaluated on the test set.

Since each scenario is treated as a ranking task over candidate models or metrics, we assess the quality of our adaptive method using Normalized Discounted Cumulative Gain (NDCG). For model selection, we use model accuracy as the NDCG relevance score; for metric selection, we use AU-ROC. To ensure robustness, we repeat the train-test split with different random seeds across 5 runs and report the average NDCG score.

Table 2: **NDCG score (NDCG@all) across three evaluation scenarios.** GAIN indicates the percentage improvement over the random selection baseline.

| Dataset | Scenario 1 | | | | Scenario 2 | | | | Scenario 3 | | | |
|---|---|---|---|---|---|---|---|---|---|---|---|---|
| | Ours | Worst | Random | Gain | Ours | Worst | Random | Gain | Ours | Worst | Random | Gain |
| CommonsenseQA | **0.973** | 0.840 | 0.933 | +4.1% | **0.878** | 0.673 | 0.846 | +3.8% | **0.967** | 0.837 | 0.930 | +3.9% |
| Math | **0.954** | 0.795 | 0.924 | +3.3% | **0.881** | 0.640 | 0.825 | +6.8% | **0.951** | 0.789 | 0.919 | +3.5% |
| TriviaQA | **0.968** | 0.826 | 0.932 | +3.8% | **0.986** | 0.919 | 0.955 | +3.2% | **0.965** | 0.822 | 0.929 | +3.9% |
| TruthfulQA | **0.969** | 0.863 | 0.936 | +3.6% | **0.811** | 0.564 | 0.751 | +8.0% | **0.965** | 0.856 | 0.933 | +3.4% |
| Average | **0.966** | 0.831 | 0.931 | +3.7% | **0.899** | 0.699 | 0.844 | +5.5% | **0.962** | 0.826 | 0.928 | +3.7% |

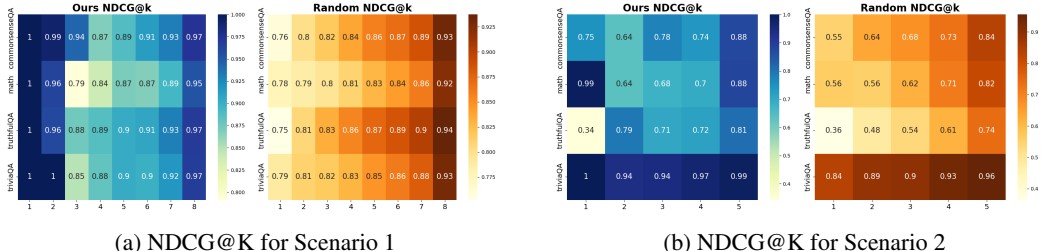

(a) NDCG@K for Scenario 1       (b) NDCG@K for Scenario 2

Figure 4: **Performance at different cutoff ranks.** Heatmaps show NDCG@K values for (a) Scenario 1 and (b) Scenario 2 of our method and random selection. Our method performs well at lower cutoff ranks (e.g., $K = 1$ or $K = 2$), but experiences a gradual decline at higher ranks, reflecting a diminishing ability to maintain relevance as more items are considered.

## 6.3 EXPERIMENT RESULT AND ANALYSIS

To evaluate our method, we compare it against two baselines: the worst-case selection and the the expected performance from random selection, computed as the average NDCG score across all candidates. This second baseline represents the score one would expect to achieve on average when selecting a metric or model at random.

As shown in Table 2, our uncertainty-guided selection consistently outperforms both baselines across all three evaluation scenarios. While the average performance is already high in some cases—often exceeding 90%—our method still achieves notable improvements. It yields average gains of 3.7% in both Scenario 1 and Scenario 3, and a larger gain of 5.5% in Scenario 2. These results demonstrate the robustness and practical benefit of using uncertainty profiles to guide metric and model selection.

Heatmaps in Figure 4 show NDCG scores at different cutoff ranks $K$ for our method and random selection in Scenarios 1 and 2. Our method excels at lower $K$, indicating it can effectively rank the most relevant items at the top. However, its performance declines at higher $K$. This is likely due to two factors: mismatches between uncertainty indicators and actual correctness and the diminishing impact of uncertainty at higher $K$ due to less distinguishable items. While our method doesn't optimize all ranks perfectly, it consistently provides meaningful improvements without requiring additional training or supervision. This makes it a practical alternative to the manual tuning process commonly used for selecting metrics or models in real-world applications.

## 7 CONCLUSION

In this study, we introduce a framework that decomposes LLM response uncertainty into four interpretable causes, each measured using specialized indicators. By profiling these uncertainty causes across different tasks, models, and uncertainty metrics, we reveal meaningful patterns that support more informed decisions in uncertainty-aware model and metric selection. Our adaptive selection approach, driven by these uncertainty profiles, consistently outperforms non-adaptive baselines. This work contributes to more interpretable, task-sensitive evaluations of LLM behavior and offers practical insights for enhancing LLM performance in real-world applications.

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

## 8 APPENDIX

### 8.1 USE OF LARGE LANGUAGE MODELS

As this work specifically investigates LLM uncertainty, the use of large language models is intrinsic to the research methodology. We employ open-source models (e.g., LLaMA-3, Gemma-2) as subjects of study within our uncertainty profiling framework, generating responses and constructing uncertainty indicators from their outputs. In addition, we used an LLM-based assistant (OpenAI GPT-5) for minor writing support, including grammar checking and improving manuscript readability. The authors paid careful attention to ensuring that AI generated content is accurate and aligned with the author's intentions.

### 8.2 DETAILS OF $\hat{SS}$ & $\hat{SA}$

In our following experiments. For $\hat{SA}$, we use the hidden state of the model last layer as semantic embedding and use cosine distance as the semantic distance metric. For $\hat{SS}$, we choose Rouge-L as the lexical distance measure.

### 8.3 Normalization and Scaling

To ensure that all four uncertainty indicators are comparable and interpretable within a unified $[0, 1]$ range, we apply appropriate scaling functions to each raw metric. $\hat{SS}$, based on lexical distance, is naturally bounded between 0 and 1 and requires no additional scaling. $\hat{SA}$, measured as cosine distance, is divided by 2 to ensure its values fall within the desired range. $\hat{IO}$, derived from entropy, is scaled using the transformation $1 - 2^{-IO}$ to compress its potentially unbounded values while preserving interpretability. $\hat{UC}$, computed as Jensen-Shannon divergence, is normalized by dividing by $\log 2$, which is its theoretical maximum. This normalization facilitates cross-cause comparison and improves the stability of downstream evaluation and visualization.

### 8.4 Uncertainty Profile Vector Conversion

For each scenario, since we rank a set of model or metric vectors based on a given vector (or vectors), the relative differences between them matter more than their absolute values. Therefore, we apply column-wise min-max scaling to convert the original uncertainty cause values into relative values before computing cosine similarity. This normalization allows us to focus purely on the relative alignment between vectors.

### 8.5 Datasets and Models

We profile uncertainty using four diverse datasets: MATH (mathematical reasoning) (Hendrycks et al., 2021), CommonsenseQA (commonsense reasoning) (Talmor et al., 2018), TRIVIAQA (reading comprehension) (Joshi et al., 2017), and TruthfulQA (safety and truthfulness) (Lin et al., 2021). These datasets are evaluated using five open-source language models of varying sizes: LLaMA-3.2-3B, LLaMA-3-8B, Gemma-2-2B, Gemma-2-9B, and Mistral-7B-v0.3. This selection enables broad coverage of task types and architectural characteristics.

### 8.6 Sampling Configuration

For metric evaluation, we follow the original configurations from each uncertainty metric's source paper—including LLM prompts and temperature settings—to ensure faithful reproduction and fair comparison. Specifically, we generate 32 samples for all *Generation Likelihood-based*, *Lexical Consistency-based*, and *Reasoning Step-based* metrics, and 5 samples for *Verbalized Confidence-based* metrics. For *EU-AU-based* metrics, we sample 5 paraphrased variants for SPUQ-COMP; and 4 iterative chains (depth 5) for IPT-EU. For SE, we use JINAAI/JINA-EMBEDDINGS-V3 as the embedding model for semantic clustering. For LLM output sampling, we intentionally use simple, straightforward, and clear instructions—following prompt styles and sampling temperatures similar to (Zhao et al., 2025)—to minimize prompt-induced variability. This controlled setup helps mitigate confounding effects from prompt design, allowing us to focus on evaluating the core behavior of each uncertainty metric under standardized conditions. All experiments are conducted on two NVIDIA GeForce RTX 3090 GPUs. We use the vLLM engine for efficient inference and Hugging Face models to encode text into vectors.

### 8.7 Statistical Significance and Evaluation Protocol

In our experiments, we sample 150 questions per dataset due to computational constraints. To improve robustness and statistical reliability, we apply bootstrapping with 500 resamples across all evaluations. Statistically, Hoeffding's inequality shows that the 150-question subsample is guaranteed to lie within $\pm 11$ percentage points of the true accuracy 95% of the time, while the data-driven bootstrap narrows this to $\pm 8$ pp ($p = 0.32$ versus the full set), confirming that our conclusions remain robust despite sub-sampling.

## 8.8 PROMPT TEMPLATES

> Paraphrase the following question, without changing its meaning.
> Make sure you only output a single question only.
> Question: {q}
> Paraphrased Question:

Figure 5: Paraphrased Prompt Template

> Clarify the following question by rewriting it in a clearer, more complete form.
> If the question is ambiguous, add missing details to make it understandable.
> Make sure you only output a single question only.
> Original Question: {q}
> Clarified Question:

Figure 6: Clarified Prompt Template

> Please answer the following question. Think carefully and in a step-by-step fashion.
> At the end of your solution, indicate your final answer by writing the answer choice (A, B, C, D, or E) inside a boxed environment, like: $\boxed{A}$.
> Q: {q}
> Choices: {c}
> Your answer:

Figure 7: Sampling Prompt Template for MC Questions

> Following is your previous response to the question.
> Q: {q}
> Choices: {c}
> Your previous response: {a}
> Check your previous response carefully and solve the same question again.
> At the end of your solution, indicate your final answer by writing one of the answer choice (only letter : A, B, C, D, or E) inside a boxed environment, like: $\boxed{A}$.
> Output:

Figure 8: Check Prompt Template for MC Questions

> Read the following passage and answer the question.
> Passage : {p}
> Question : {q}
> At the end of your solution, indicate your final answer inside a boxed environment, like: $\boxed{answer}$.

Figure 9: Sampling Prompt Template for RC Questions

Following is your previous response to the question:
Read the following passage and answer the question.
Passage : {p}
Question : {q}
Your previous response: {a}
Check your previous response carefully and respond the question again.
At the end of your solution, indicate your final answer inside a boxed environment, like: $\boxed{answer}$.

Figure 10: Check Prompt Template for RC Questions

Please answer the following question.
Think carefully and in a step-by-step fashion.
At the end of your solution, put your final result in a boxed environment, e.g. $\boxed{answer}$.
Q: {q}

Figure 11: Sampling Prompt Template for Essay Questions

Following is your previous response to the question.
Q: {q}
Your previous response: {a}
Check your previous response carefully and solve the same question again step by step.
At the end of your solution, put your final result in a boxed environment, eg. ($\boxed{answer}$).
Output:

Figure 12: Check Prompt Template for Essay Questions

