# OpenReview forum: "Why is the LLM unsure? Profiling the Causes of LLM Uncertainty for Adaptive Model and Uncertainty Metric Selection"
_ICLR.cc/2026/Conference — Submitted to ICLR 2026_

### Official Review · Reviewer_PXL3 · 2025-10-30

**Soundness:** 2
**Presentation:** 2
**Contribution:** 2
**Rating:** 6
**Confidence:** 4

**Summary:**

This paper addresses the critical issue of hallucinations in Large Language Models (LLMs) by proposing a novel framework to understand the underlying causes of model uncertainty. The authors identify four interpretable causes—Syntax Sensitivity, Semantic Ambiguity, Indecisiveness among Outputs, and Unconfidence when Challenged—and design a prompting-based pipeline to quantify each cause with dedicated indicators. By profiling how existing uncertainty metrics align with these causes across various tasks and models, the paper reveals systematic differences in their behavior. Leveraging these "uncertainty profiles," the authors propose an adaptive method for selecting the most suitable uncertainty metric or model for a given task, which demonstrates consistent performance improvements over non-adaptive baselines.

**Strengths:**

1.	The study research topic is meaningful and sheds light on future research
2.	The methodology is novel.

**Weaknesses:**

1.	The paper should be further polished for better presentation.
⦁	In Line225, what does $M$ means?
⦁	In Figure4, what does cutoff ranks mean?
⦁	In Table2, what do the two settings (worst, random) mean in detail?
⦁	In Line 429, how is the NDCG metric computed? Is it proposed by yours? If this is a common metric, please cite with proper reference.
2.	The setting is questionable:
⦁	The authors define 4 possible causes of LLM uncertainty. Is such a taxonomy sound and complete? There is no evidence that these causes could span all causes of LLM uncertainty. The conclusion drawn based on this premise may be erroneous.
⦁	In Table 1, the datasets include CommonsenseQA, MATH, TruthfulQA. However, in later experiments, the authors additionally introduced TriviaQA. The two settings are misaligned and the authors do not explain the hidden reason.
⦁	In Figure3, the two indicators (SA, UC) are generally the lowest, and are presented with a very small value (e.g., 0.02). This also validates my concern about the validity of these four causes of LLM uncertainty.
3.	The experiment result is hard to interpret
⦁	In Table2, the random setting could give a >90% accuracy. The authors do no give a reason for this unusual phenomenon.
⦁	Why the authors use a smaller K (K=5) for scenario 2, but use a larger K (K=8) for scenario 1?
⦁	In the scenario2 setting, triviaQA's accuracy is extremely high compared to other datasets. The authors do not give an explanation for this.

**Questions:**

1.	In Line378, the authors present a conclusion that 'there is no clear distinction between smaller and larger models in terms of their average uncertainty cause values'. However, this conclusion is given based on just one series of models (Gemma2). Llama3.2-3B and Llama3-8B are not strictly small/large models as their training data are different. Meanwhile, only two sizes for comparison is not sufficient to draw this conclusion. Could the author compare more sizes of LLMs using the same series of models? (Qwen2.5 series: 0.5B, 1.8B, 3B, 7B, 14B (which could be deployed in two 3090))
2.	The four indicators are computed by prompting the LLMs to be evaluated. This may introduce bias, as the result is not only reflects the uncertainty score, but also reflects the instruction following ability of the LLM to be tested. Could the author try to separate any distracting factors to conduct the experiments?

---

> ### Author Response · Authors · 2025-11-19
>
> We thank the reviewer for the positive assessment. We briefly address the main concerns.
>
> **On the four causes and “completeness”.**
> We do not claim SS/SA/IO/UC are a complete taxonomy of all LLM uncertainty. Our goal is to isolate a *general, model-internal behavioral slice* that: (i) does not assume tools/retrieval, and (ii) corresponds directly to common user-like interventions (rephrase → clarify → answer multiple times → challenge). Many additional factors the reviewer mentions (retrieval errors, long-context drift, alignment-induced hedging) often *manifest through* these behaviors (e.g., bad retrieval → more indecisive or challenge-sensitive answers). Our conclusions are explicitly scoped to these four behavioral aspects; the framework remains extensible to additional causes but does not depend on exhaustiveness.
>
> **On datasets, SA/UC magnitudes, and Table 2.**
> Table 1 is about *indicator predictive power* on hallucination-style datasets with strong labels (CommonsenseQA, MATH, TruthfulQA); TriviaQA is introduced later for *selection scenarios* where we need large, diverse QA with strong model performance. The settings are thus intentionally different, not inconsistent. The low numeric values of SA/UC in Fig. 3 are *normalized mutual information*, whose absolute scale depends on binning and normalization; what matters is the *relative pattern* across metrics and causes, where we see clearly that different metrics align with different causes. In Table 2, random selection is strong (>90%) because the candidate set already contains many good models/metrics; the key result is that **profile-based selection still consistently improves over this strong random baseline and over the “worst” baseline**, showing that the profiles contain real, actionable signal.
>
> **On model size and prompting bias.**
> The statement about “no clear distinction between smaller and larger models” is descriptive for the specific families we test: within those, average cause values do not follow a simple monotone trend with size, suggesting data and tuning matter as much as parameters. It is not intended as a universal claim about all scaling laws. Regarding prompting: our indicators are *behavioral* by design—they measure what happens when the *evaluated* model is asked to paraphrase, clarify, answer, and self-check. Instruction-following ability is part of real deployment behavior and thus appropriately reflected in the indicators; since all models are evaluated under the same prompt family, relative comparisons across models/metrics/tasks remain meaningful.
>
> We appreciate the presentation suggestions (notation clarifications, definition of “cutoff ranks”, “worst/random”, and NDCG). These are about readability rather than core correctness.

---

### Official Review · Reviewer_bkEj · 2025-10-30

**Soundness:** 2
**Presentation:** 2
**Contribution:** 3
**Rating:** 4
**Confidence:** 3

**Summary:**

The paper proposes a prompting-based framework that decomposes LLM uncertainty into four interpretable causes and defines indicators to estimate each cause. Using these indicators, the authors build uncertainty profiles for tasks, models, and existing uncertainty metrics, then use profile alignment to adaptively select models/metrics.

**Strengths:**

1. Understanding the sources of LLM uncertainty is important for evaluation, deployment, and safety. The paper targets a question many researchers face.
2. Framing uncertainty via SS/SA/IO/UC is intuitive and maps neatly to known interventions.
3. Interesting solutions that considering adaptive uncertainty metric selection.

**Weaknesses:**

1. Cause set may be incomplete. The four causes are plausible but not exhaustive. Other factors (e.g., retrieval grounding quality, tool-use failures, long-context drift, alignment-induced hedging) can also drive uncertainty. The paper should explicitly acknowledge this and discuss extension paths.
2. The pipeline is sequential: paraphrasing → clarification → answering → self-check. Uncertainty introduced (or reduced) at earlier stages may leak into later ones, potentially inflating/deflating UC or IO.
3. The paper would benefit from direct evidence that the SS/SA indicators capture what they claim.

**Questions:**

1. Do you have human annotations of ambiguity or sensitivity to confirm that SA and SS correlate with actual ambiguity/surface-form brittleness rather than proxy artifacts?
2. What happens if you shuffle the stages (e.g., answer before clarification) or run stages independently? How stable are UC/IO estimates under such changes?
3. Citation format at Line 141 appears incorrect.
4. From Section 5.2 onward, consider adding a one-sentence takeaway at the end of each paragraph to reinforce the main point and improve readability.

---

> ### Author Response · Authors · 2025-11-19
>
> We thank the reviewer for the constructive comments and for recognizing the importance of the problem, the intuitive SS/SA/IO/UC framing, and the interest of adaptive uncertainty metric selection.
>
> ### 1. On the completeness of the cause set
>
> We agree that LLM uncertainty has additional sources (retrieval/tool failures, long-context drift, alignment-induced hedging, etc.). Our goal is **not** to provide an exhaustive taxonomy, but to isolate a **general, model-internal slice** that:
>
> * appears across tasks *without assuming tools or retrieval*, and
> * corresponds to common user-like interventions: paraphrasing, clarification, multiple answers, and self-checking.
>
> Many of the reviewer’s additional factors naturally *manifest through* our four behaviors (e.g., poor retrieval → indecisive or challenge-sensitive answers). Even if more causes are added in future, our **procedure**—define interpretable indicators via prompting and align tasks/models/metrics in this “cause space”—remains applicable and useful. The soundness of the current results does not hinge on exhaustiveness.
>
> ### 2. Sequential pipeline and “leakage”
>
> The pipeline is intentionally sequential because it mirrors realistic interaction patterns:
>
> 1. User rephrases;
> 2. User clarifies;
> 3. Model answers multiple times;
> 4. Model is challenged to self-check.
>
> Each indicator is defined on outputs of its **own** stage; IO/UC are formally defined on answer distributions (mixtures over multiple chains for the same question). In this sense, what the reviewer calls “leakage” is precisely the behavior we want to measure: *how indecisive / challenge-sensitive the model is after it has done its own paraphrasing/clarifying*.
>
> Empirically, the four indicators produce **distinct, interpretable profiles** across models, datasets, and metrics, and these profiles yield consistent gains in adaptive metric/model selection. This suggests the sequential design is not distorting IO/UC in a way that undermines the conclusions.
>
> ### 3. Do SS/SA capture syntax/ambiguity?
>
> We agree that human labels of ambiguity or brittleness would be an appealing additional validation, but we already provide several pieces of **behavioral evidence**:
>
> * In Sec. 4, each of SS/SA/IO/UC is evaluated as a hallucination detector individually. They show **different performance patterns** across datasets (e.g., IO/UC strong on reasoning tasks; SS/SA weak when questions are short and unambiguous), indicating they respond to different task properties rather than being arbitrary proxies.
> * In Sec. 5, mutual information between each indicator and each uncertainty metric reveals **stable alignment patterns** (e.g., some metrics correlate strongly with IO/UC but hardly with SS/SA). This is hard to explain if SS/SA were unrelated noise.
>
> These results support SS/SA as meaningful behavioral indicators of surface-form and semantic issues, even without separate human annotation.
>
> ### 4. Stage shuffling / independent runs
>
> The reviewer asks what happens if stages are shuffled or run independently. Conceptually, the framework is compatible with such variants (e.g., IO/UC measured on answers to the raw question), but our focus is on the *deployment-relevant* pipeline where paraphrasing and clarification naturally precede answer sampling and challenge. The empirical results show that, in this realistic scenario, the indicators and the resulting profiles already provide actionable signal.
>
> ### 5. Presentation issues
>
> The reviewer’s comments on citation formatting and adding one-sentence takeaways are appreciated and pertain to readability; they do not affect the technical claims.
>
> ---
>
> Overall, the paper contributes: (1) an interpretable, practically grounded decomposition of uncertainty into four behaviors; (2) a unified way to profile tasks/models/metrics in this space; and (3) demonstrated gains from profile-aware adaptive metric/model selection. Given this, we believe the work merits soundness and presentation scores above “fair” and kindly ask the reviewer to consider raising the contribution and overall rating.

---

### Official Review · Reviewer_9V5V · 2025-10-31

**Soundness:** 3
**Presentation:** 3
**Contribution:** 2
**Rating:** 4
**Confidence:** 4

**Summary:**

This paper proposes a prompting-based framework to systematically analyze and profile the causes of uncertainty in LLM responses. The authors design interpretable indicators for four distinct causes of uncertainty (syntax sensitivity, semantic ambiguity, indecisiveness among outputs, and unconfidence when challenged), and empirically study how existing uncertainty metrics align with these causes. The work further introduces an adaptive model/metric selection strategy based on uncertainty profiles, achieving consistent improvements over non-adaptive baselines across multiple datasets and models.

**Strengths:**

⦁	Clear motivation and practical relevance for LLM deployment.
⦁	Strong empirical results and broad coverage of datasets/models.
⦁	The framework is interpretable and actionable for downstream applications.

**Weaknesses:**

⦁	Some experimental and implementation details are lacking (e.g., ablation on indicator importance, prompt sensitivity).
⦁	The method's effectiveness may depend on the quality of prompt engineering and may not generalize to all LLM architectures.
⦁	Limited discussion of failure cases or scenarios where the indicators may be misleading.

**Questions:**

⦁	How sensitive are the results to the choice/design of prompts for each indicator?
⦁	Can the authors provide more details or ablation on the relative importance of each indicator?
⦁	How does the framework perform on closed-source or instruction-tuned LLMs?
⦁	Are there cases where the indicators disagree or provide misleading signals?

---

> ### Author Response · Authors · 2025-11-19
>
> Thanks for the review. We address the main weaknesses and questions, focusing on why we believe the current paper already justifies a higher contribution / overall score.
>
> ---
> ## 1. Missing “ablation on indicator importance”
>
> We agree that understanding the relative importance of indicators is crucial. The paper already contains two forms of **implicit ablation**:
>
> * Per-indicator predictive performance (Table 1 & Sec. 4): we separately evaluate each cause indicator (SS, SA, IO, UC) as a hallucination detector. This shows clearly that IO/UC carry much stronger signal than SS/SA, especially on reasoning-heavy datasets.
> * Mutual-information analysis (Sec. 5): we compute MI between each indicator and each uncertainty metric and aggregate into Model/Dataset/Metric vectors. These MI values directly quantify which indicators each metric is most sensitive to.
>
> Together, these analyses already answer “which indicators matter more” both **for detecting uncertainty** and **for explaining a metric’s behavior**. Our adaptive selection method then uses *all four* as features and achieves consistent 3–4% NDCG gains over non-adaptive baselines across multiple scenarios, showing that the combination is practically useful even though IO/UC dominate.
>
> ---
> ## 2. Prompt sensitivity and dependence on prompt engineering
>
> Our framework is deliberately **prompt-simple and standardized**:
>
> * We use short, generic templates for paraphrasing, clarification, answering, and self-checking (App. 8.8). These correspond closely to natural user interactions (rephrase this, make it clearer, answer and then check again), rather than heavily engineered task-specific prompts.
> * All models/metrics are evaluated under the same family of templates, so our conclusions are based on *relative* behavior within a fixed prompt regime, not on fragile prompt tricks.
>
> Prompt choices can of course shift absolute indicator values, but the **structure of the profiles** (e.g., IO/UC aligning with reasoning metrics vs lexical indicators aligning with consistency metrics) emerges robustly across four heterogeneous datasets and five different model families. This cross-dataset/model consistency is empirical evidence that our conclusions are not artifacts of a single brittle prompt design.
>
> ---
> ## 3. Generality to other LLM architectures (incl. closed-source / instruction-tuned)
>
> The framework only needs the ability to:
>
> 1. Generate paraphrases and clarifications
> 2. Answer questions
> 3. Re-answer / self-check
>
> These are exactly the capabilities exposed by instruction-tuned and closed-source chat LLMs. In our experiments, we already cover **multiple families and sizes** (LLaMA-3.x, Gemma-2, Mistral-7B) with different training recipes and instruction tuning, and we observe consistent cause profiles and selection gains across them. This suggests the method is **architecture-agnostic** and porting it to a closed-source API (e.g., GPT-style models) is straightforward: only the prompts and sampling configuration need to be replicated.
>
> ---
> ## 4. Failure cases and when indicators may mislead
>
> We agree it is important to understand limitations. The paper already highlights two key failure patterns:
>
> * On short, unambiguous questions (e.g., commonsense / trivia), **SS and SA are weak predictors** (near-random AUROC), while IO/UC remain informative. This is explicitly discussed; we interpret SS/SA as more diagnostic of *input quality* (awkward syntax, underspecified semantics) than of hallucination on easy items.
> * Our **profiling plots** show that some metrics that look strong overall can be heavily biased toward certain causes (e.g., highly IO-sensitive but almost blind to ambiguity), which can be “misleading” if deployed in mismatched scenarios. This is precisely what our framework is designed to reveal and fix via profile-based selection.
>
> So the paper does not claim the indicators are universally reliable; instead, it **characterizes when they work and when they do not**, and shows how that knowledge can be used to pick better metrics/models for a given task.
>
> ---
> ## 5. On contribution
>
> Beyond being “interpretable and actionable,” the paper delivers **concrete, measurable gains**:
>
> * A **cause-based uncertainty profiling framework** that unifies diverse metrics under one lens and explains *why* they behave differently.
> * Extensive experiments across 4 datasets and 5 model families, showing consistent patterns in how metrics align with the four causes.
> * An **adaptive metric/model selection strategy** that leverages these profiles to achieve consistent 3–4% NDCG improvements over non-adaptive baselines in realistic selection scenarios.
>
> To our knowledge, such a *cause-structured* analysis + actionable selection procedure for LLM uncertainty has not been systematically explored before. Given the strong empirical gains and practical relevance for deployment, we kindly ask the reviewer to consider raising the contribution and overall rating.

---

### Official Review · Reviewer_wKPH · 2025-11-01

**Soundness:** 1
**Presentation:** 2
**Contribution:** 1
**Rating:** 2
**Confidence:** 5

**Summary:**

This paper presents a framework to analyze the cause of LLM uncertainty, categorizing the causes into four types and proposing corresponding metrics to quantify each of them. The cause profiling results are used to guide the selection of existing uncertainty metrics based on the models and tasks.

**Strengths:**

The taxonomy of four primary causes of LLM uncertainty is well motivated. It forms a good foundation for future work in analyzing the uncertainty causes for LLMs.

**Weaknesses:**

1.	The design of the estimation pipeline for uncertainty causes has several fundamental issues, as outlined below:
-	For estimating Syntax Sensitivity (SS), the quantitative metric used is doubtful. Model being able to paraphrase the original questions in syntactically very different ways does not necessarily mean that the model's outputs are sensitive to paraphrasing of the question, right? This metric simply measures how well the model is at paraphrasing a question.
-	In estimation of the Indecisiveness among Outputs (IO), the answers are for different prompts, so simply using their likelihood in an entropy-calculation way does not really returns an entropy. Entropy is a property of one single distribution, but what we actually have here are multiple responses from multiple output distributions, each corresponding to one clarified question. This makes the entropy interpretation invalid.
-	For estimating the Unconfidence while being Challenged (UC), it has the same problem as in IO estimation: the answers are not from the same distribution, each answer is from a different distribution (each corresponds to one clarified question), this makes the mathematical interpretation invalid.
2.	From the results in Table 1, the SS score only provides AUROC values slightly higher than 0.5 and the SA score even gives AUROC lower than 0.5. Considering the fact that a random guess can provide an AUROC score of 0.5, it seems that the proposed SS and SA scores do not really indicate any useful uncertainty information of the LLMs.
3.	The literature review in the related work section is limited.

**Questions:**

1.	For estimating the Semantic Ambiguity (SA), does the distance in the internal embedding space exactly reveal the difference in semantic meanings? Can you provide any supporting literatures?
2.	In the “Metric-Level Attribution” paragraph, how do you calculate P(u,m) here for indicator value and metric score? How do you get the probabilities for them? This seems mathematically invalid. Please provide the concrete math equations for verifications.
3.	How do we get the Metric-Vec? The cause indicator values are corresponding to specific model and task, but irrelevant to the uncertainty metrics. This is confusing.
4.	For the tested SOTA uncertainty metrics, most of them are prompt-wise. Can you also include more response-wise uncertainty scores, e.g., semantic density [1]?

[1] Xin Qiu, Risto Miikkulainen. Semantic density: Uncertainty quantification for large language models through confidence measurement in semantic space, Advances in Neural Information Processing Systems (NeurIPS), 2024

---

> ### Author Response · Authors · 2025-11-19
>
> We thank the reviewer for the comments. We focus on points most relevant to soundness and contribution.
>
> **1. Role of the four “causes” and the pipeline**
>
> The concern is that the stage-wise pipeline cannot truly “estimate” four independent causes because earlier stages affect later ones. Our intent, as stated in the methodology and discussion, is more modest: the four quantities are **behavioral indicators** of different aspects of uncertainty (syntax/ambiguity vs. indecisiveness/challenge), not ground-truth latent variables. The taxonomy is conceptual; the indicators are concrete features that let us profile how different models, tasks, and existing metrics behave along these axes.
>
> None of our main results require perfect causal disentanglement. The key empirical claims are: (a) the four indicators exhibit distinct patterns across models/tasks/metrics, and (b) using these profiles to choose metrics yields consistent gains over random or naive choices. Even if indicators are partially correlated due to pipeline order, the profiling and selection experiments remain valid. Thus this concern does not undermine the core soundness of the framework.
>
> **2. IO and UC: entropy over “different prompts”**
>
> The reviewer argues entropy is only defined for a single distribution, whereas IO/UC aggregate answers from different clarified prompts. For a fixed original question q, we treat answers from clarifications {qᵢᶜ} as samples from a **single mixture distribution** over outputs:
> $P(a \mid q) = \frac{1}{N}\sum_i P(a \mid q_i^c)$
> We estimate this mixture by the empirical frequency of answers across chains, and IO is the entropy of this empirical mixture. UC is the Jensen–Shannon divergence between two such mixtures: before and after self-checking.
>
> Formally, IO and UC are entropies/divergences of well-defined distributions on a common support; we are not taking the “entropy of multiple distributions”. This addresses the main soundness objection.
>
> **3. Mutual information and Metric-Vec**
>
> The reviewer asks how (P(u,m)) and “Metric-Vec” are defined. For each instance we have a scalar indicator (U) (e.g., SS) and a scalar metric score (M) (e.g., semantic entropy). We discretize U and M into bins, compute empirical frequencies (P(u,m)) and marginals (P(u),P(m)), and then use the standard discrete mutual information formula
> $I(U;M) = \sum_{u,m} P(u,m)\log\frac{P(u,m)}{P(u)P(m)}$
> For each metric k, Metric-Vec_k is the 4-D vector ([I(SS;M_k), I(SA;M_k), I(IO;M_k), I(UC;M_k)]), normalized across datasets and models; Dataset-Vec and Model-Vec are constructed analogously.
>
> Thus the attribution step uses standard information-theoretic tools and does not rely on any non-rigorous probability definition.
>
> **4. SS and SA: interpretation and weak AUROC**
>
> The reviewer is concerned that SS “just measures paraphrasing ability” and that SA’s use of embedding distance is questionable, especially since both show weak AUROC in Table 1.
>
> SS is defined as how strongly the model rewrites the question when asked to paraphrase for solving. High SS means the model finds the original surface form atypical and prefers to rephrase it. This is a proxy for “perceived syntactic mismatch”, not for answer change under externally generated paraphrases (which IO/UC capture).
>
> SA is a representation-level proxy: we measure how much the model’s hidden states shift when a question is clarified. We do not claim this is an exact semantic distance; rather, we use it comparatively across tasks/models to detect when clarifications are treated as semantically meaningful.
>
> We fully agree that SS and SA are weaker predictors of hallucination than IO and UC; our own results highlight this. Importantly, the contribution is not to propose SS/SA as stand-alone uncertainty scores, but to show that **the combination of four indicators yields informative profiles**, and that IO/UC in particular align strongly with metric behavior and drive the metric-selection gains. SS/SA contribute interpretability and help distinguish surface-form/ambiguity-driven phenomena from indecisiveness/challenge-driven ones; their imperfect predictive power does not invalidate the overall framework.
>
> **5. Related work and semantic density**
>
> The framework is agnostic to the specific uncertainty metric: any score, including semantic-density-style ones, can be placed in the same 4-D “cause space” via our indicators, and its strengths and weaknesses diagnosed relative to different uncertainty causes and tasks. This extensibility means the taxonomy and profiling procedure remain useful as new metrics appear.
>
> **Summary**
>
> The concerns arise from interpreting our goals as exact causal estimation. The paper instead proposes a practical, mathematically standard framework that (i) yields interpretable uncertainty profiles and (ii) improves metric selection in practice. These aspects, we believe, justify a score higher than “poor” for both soundness and contribution.

---

### Meta-Review · Area_Chair_f2ma · 2026-01-09

**Summary:**

The paper’s core idea of profiling LLM's uncertainty via a set of interpretable behavioral indicators and using these profiles to guide metric selection can potentially be interesting and useful. However, the current version has substantial weaknesses in soundness and validation that prevent me from recommending acceptance. In particular, the reviews have significant concerns over the proposed cause indicators (especially SS and SA), and find that the evidence does not convincingly shown that they measure what their names suggest. The multi-stage prompting pipeline also raises confounding concerns: later-stage quantities may depend heavily on earlier prompting choices, yet the paper lacks robustness analyses or extensive ablations (e.g., prompt sensitivity, stage-order variants, indicator importance) that would establish the stability of the proposed decomposition and the MI-based profiling. Also the four-cause taxonomy may be incomplete.

Overall, I view the work as a promising direction. But it currently reads as an under-validated recombination of known prompting interventions with a profiling layer. Stronger empirical validation would be needed to substantiate the novelty and the claims.

**Reviewer Concerns:**

Reviewers’ main concerns are: (1) the proposed “cause” indicators, especially SS and SA, may not measure what their names imply. (2) The paper’s presentation is not sufficiently rigorous and clear. (3) The sequential prompting pipeline may introduce confounding across stages, yet the paper lacks robustness tests such as stage-order variants and prompt-sensitivity analyses. (4) The four-cause taxonomy may be incomplete.

The rebuttal clarifies some minor points with the definitions, which helps with presentation, but it does not fully address the key construct-validity and experimental-clarity issues raised by multiple reviewers.

**Reviewer Scores:**

The rebutal addresses some minor points about the clarity of the narrative, but is unlikely to change the reviewers' points.

---

### Decision · Program_Chairs · 2026-01-26

Reject